# Advances in the Diagnosis and Treatment of Myeloproliferative Neoplasms (MPNs)

**DOI:** 10.3390/cancers17193142

**Published:** 2025-09-27

**Authors:** Xinyu Ma, Zhibo Zhou, Shuyu Gu, Yan Guo, Tianqing Zhou, Ruonan Shao, Jinsong Yan, Wei Chen, Xiaofeng Shi

**Affiliations:** 1Department of Hematology, The Second Affiliated Hospital of Nanjing Medical University, Nanjing 210003, China; xinyuma@stu.njmu.edu.cn (X.M.); zhouzhibo@stu.njmu.edu.cn (Z.Z.); 23011833@stu.njmu.edu.cn (S.G.); guoyan2023@stu.njmu.edu.cn (Y.G.); zhoutianqing@stu.njmu.edu.cn (T.Z.); shaoruonan1999@stu.njmu.edu.cn (R.S.); 2The Second Clinical Medical School, Nanjing Medical University, Nanjing 210011, China; 3Department of Hematology, Liaoning Medical Center for Hematopoietic Stem Cell Transplantation, The Second Hospital of Dalian Medical University, Dalian 116027, China; yanjsdmu@dmu.edu.cn; 4Key Laboratory of Bone Marrow Stem Cells, Department of Hematology, Blood Diseases Institute, Affiliated Hospital of Xuzhou Medical University, No. 99, Western Huaihai Road, Xuzhou 221004, China

**Keywords:** myeloproliferative neoplasms (MPNs), polycythemia vera (PV), essential thrombocythemia (ET), primary myelofibrosis (PMF), driver mutations, risk stratification, diagnosis and treatment, challenges and future directions

## Abstract

Myeloproliferative neoplasms are rare blood disorders caused by genetic mutations that lead to uncontrolled production of blood cells, increasing risks of blood clots and other complications. Recent advances in molecular diagnostics have transformed diagnosis through identifying key mutations. Treatment now focuses on targeted therapies to reduce symptoms, prevent complications, and improve quality of life. We summarize the latest advancements in the diagnosis and treatment of myeloproliferative neoplasms, highlighting the importance of molecular mechanisms in guiding therapeutic approaches and the potential for precision medicine in the future.

## 1. Introduction

### 1.1. Background

Myeloproliferative neoplasms (MPNs) are clonal myeloid malignancies defined by bone marrow (BM) hyperplasia and peripheral blood (PB) cytosis [1]. The main subtypes of MPN include primary myelofibrosis (PMF), polycythemia vera (PV), and essential thrombocythemia (ET). Current treatment strategies primarily aim to prevent the progression of MPNs and manage symptoms, which are largely palliative. Allogeneic hematopoietic stem cell transplantation (HSCT) remains the sole curative option [2].

Since the discovery of the *JAK2 V617F* mutation in 2005, significant advances have been made in elucidating the molecular pathogenesis of MPNs. In particular, somatic gene mutations involving the *janus kinase 2* (*JAK2*), *calreticulin* (*CALR*), and *Myeloproliferative leukemia virus* (*MPL*) cause a substantial hyperactivity of the Janus kinase (JAK)-signal transducer and activator of transcription (JAK-STAT) signaling pathway. This increased understanding of the molecular principles behind MPN pathogenesis has aided the development of more sensible treatment approaches.

### 1.2. Purpose

In recent years, newly developed treatments have shown efficacy in controlling symptoms and reducing the incidence of cardiovascular events, with some therapies potentially extending survival. These advanced therapies include JAK2 inhibitors, interferons, and targeted agents in clinical trials. However, active research into therapies targeting JAK/STAT pathway activation—including JAK2 inhibitors and novel MEK/ERK kinase inhibitors—has not yet yielded capability of durably reducing mutational allele burden or preventing transformation to myelofibrosis (MF)/acute myeloid leukemia (AML).

This review summarizes recent advances in MPN diagnosis, prognosis and therapy, providing clinicians with a molecular diagnostic-management framework and identification of emerging research targets.

### 1.3. Overview of MPNs

The prevalence of MPN is between 0.5 and 2.5 cases per 100,000 for PV, 1–2.5 cases per 100,000 for ET [3], 0.5 cases per 100,000 for PMF, and 0.2 cases per 100,000 for secondary myelofibrosis (SMF) [4]. Survival outcomes among MPN patients vary significantly between disease subtypes. In a large cohort of 1581 patients, the median survival was 13.5 years for PV, 19.8 years for ET, and 5.9 years for PMF [5].

The current subclassification of MPNs is based on alterations in blood cell counts and hyperplasia and dysplasia in hematopoietic lineages in the BM. The clinically evident types include PV, ET, and PMF, whereas the subclinical types are more complex. The International Consensus Classification (ICC) divides PMF into two phases: “pre-fibrotic” and “overtly fibrotic.” Furthermore, about 15% of individuals with ET or PV may eventually experience a PMF-like phenotype; this condition is known as SMF, post-ET MF, or post-PV MF [6]. The treatment and outcomes for these syndromes resemble those of PMF [7].

## 2. Pathogenesis and Molecular Mechanism

Research on MPNs has established that driver mutations, epigenetic dysregulation, and signaling pathway abnormalities collectively determine pathogenesis.

### 2.1. Clonal Hematopoiesis as a Precursor to MPNs

In recent years, clonal hematopoiesis (CH), particularly its most common form, clonal hematopoiesis of indeterminate potential (CHIP), has been recognized as a critical precursor state to the development of MPNs. CHIP is characterized by age-related somatic driver mutations (e.g., *JAK2 V617F*, *CALR*, or *MPL* mutations) leading to clonal expansion of hematopoietic stem cells, with a variant allele frequency (VAF) ≥ 2% in peripheral blood, but without meeting the diagnostic criteria for hematologic malignancies [8]. Studies show that individuals with CHIP harboring *JAK2 V617F* mutations have a significantly higher annual risk of progressing to overt MPNs compared to the general population. The pathogenesis of MPNs can be viewed as a multistep clonal evolution process built upon CHIP [9]. Initial “driver mutations” confer a growth advantage to clones, establishing the foundation for CH. Subsequently, “secondary hits”—such as non-driver mutations in *TET2*, *ASXL1*, *DNMT3A*, and epigenetic alterations—further drive the evolution and expansion of malignant clones [10].

Concurrently, interactions between mutated cells and the inflammatory microenvironment lead to persistent dysregulation of signaling pathways, such as JAK-STAT, ultimately overcoming the body’s homeostatic regulatory mechanisms and progressing to MPNs that meet diagnostic criteria [11]. Thus, MPNs essentially represent an advanced, symptomatic state of clonal hematopoiesis [12]. Understanding the progression trajectory from CHIP to MPN is crucial for early identification of high-risk individuals and developing intervention strategies.

### 2.2. Driver Mutations

The *JAK2 V617F* mutation constitutes the principal oncogenic driver, occurring in 95% of PV and 50–60% of ET or PMF cases [13]. This molecular lesion originates from a c.1849G>T substitution in exon 14, replacing valine with phenylalanine at position 617 (V617F) within JAK2’s pseudokinase domain (JH2). The mutation disrupts JH2-mediated autoinhibition of the kinase domain (JH1), permitting constitutive cytokine-independent activation [14]. Persistently activated JAK2 phosphorylates STAT3/STAT5 transcription factors, inducing their nuclear translocation and binding to gamma-interferon activation sites (GAS) within target gene promoters. Consequently, this process upregulates pro-proliferative genes (including *MYC* and *cyclin D1*), anti-apoptotic genes (such as *BCLXL*), and lineage-specific transcription factors (*GATA1, NF-E2*), ultimately driving dysregulated hematopoiesis [15] (Figure 1 and Figure 2). In PV, JAK2-STAT5 hyperactivation specifically stimulates erythroid progenitor cells, causing pathological hemoglobin overproduction. Conversely, in ET and PMF, STAT3-dependent thrombopoiesis and TGF-β/PDGF secretion mediate thrombocytosis and bone marrow fibrosis (BMF) [15]. Mutant clones further remodel the BM niche through sustained secretion of pro-inflammatory cytokines (IL-6, TNF-α), establishing a self-amplifying inflammatory circuit. This microenvironment fosters acquisition of secondary epigenetic mutations (e.g., *ASXL1*, and *TET2*), culminating in therapeutic resistance or leukemic transformation [16].

Additional driver mutations include *MPL W515L/K* variants, which disrupt the thrombopoietin receptor’s (TPO-R) autoinhibitory conformation, inducing ligand-independent JAK2-STAT5 pathway activation and consequent megakaryocytic hyperplasia [15,17]. Similarly, *CALR* type 1 or type 2 frameshift mutations generate novel C-terminal domains that aberrantly bind and activate TPO-R. This activation engages the same signaling axis while uniquely promoting abnormal membrane localization and inflammatory factor release, further exacerbating fibrotic processes [17].

### 2.3. Epigenetic Changes

Epigenetic dysregulation forms a dynamic network interacting with driver mutations and inflammatory signaling. DNA methylation abnormalities constitute a major component. *Ten-eleven translocation* (*TET2)* mutations (present in 10–20% of MPNs) impair the conversion of 5-methylcytosine(5-MC) to 5-hydroxymethylcytosine(5-HMC), leading to pathological promoter hypermethylation. This epigenetic change silences tumor suppressor genes (e.g., *SOCS1*) and affects critical differentiation regulators (e.g., *C/EBPα*) [18]. DNA methyltransferase *(DNMT)-3A* mutations conversely induce genome-wide hypomethylation, activating oncogenic pathways such as Wnt/β-catenin and enhancing clonal fitness [19]. Notably, *TET2* lesions frequently precede driver mutations, conferring hematopoietic stem cell (HSC) proliferative advantages during the CHIP phase, thereby establishing a pre-malignant foundation [20].

Histone modification defects represent another critical layer. Additional sex combs like 1 (ASXL1) through its PHD domain interacts with polycomb repressive complex 2 (PRC2) which catalyzes trimethylation (Me3) of H3 on lysine 27. *ASXL1* mutations (occurring in ~30% of PMF cases) cause a truncation of PHD domain from ASXL1, disrupting the interaction with PRC2, which reduces the level of trimethylation of H3 on lysine 27. This results in expression of profibrotic genes (*TGF-β1*, *COL1A1*) and accelerating BMF [21]. Enhancer of zeste homolog (*EZH2)* mutations (e.g., Y646F), affecting a core PRC2 component, similarly compromise H3K27me3 levels, leading to aberrant silencing of differentiation-associated genes like *HOXA9* and maintenance of stemness [22] (Figure 3). MicroRNA dysregulation further amplifies pathological signaling. Reduced *microRNA-146a* (*miR-146a*) expression activates TLR/NF-κB signaling, escalating pro-inflammatory cytokine (IL-6, TNF-α) emission and microenvironmental inflammation. Conversely, *microRNA-155* (*miR-155*) overexpression inhibits SHIP1, augmenting PI3K/AKT-mediated metabolic reprogramming, cellular proliferation, and drug resistance [23].

Significant synergism exists between epigenetic defects and driver mutations. Co-occurrence of *TET2* loss and *JAK2 V617F* suppresses *Pu.1* expression, impairing erythroid differentiation [24]. Similarly, *ASXL1* mutations cooperate with *CALR* mutations to potentiate TGF-β pathway activation, driving fibrosis [25]. Therapeutically, hypomethylating agents like azacitidine can partially restore *SOCS1* expression. However, 2-hydroxyglutarate (2-HG) accumulation resulting from *IDH* mutations may antagonize efficacy. Furthermore, *EZH2* inhibitors demonstrate limited activity in patients harboring *ASXL1* mutations [26]. In the future, it will be necessary to analyze the apparent spatiotemporal dynamics in combination with single-cell multiomics, develop precise interventions, and reverse the malignant trajectory of MPN.

### 2.4. Dysregulation of Signaling Pathways

Abnormal stimulation of the JAK-STAT signaling cascade is the core mechanism of MPN pathogenesis and is driven mainly by *JAK2 V617F*, *CALR* or *MPL* mutations. The *JAK2 V617F* mutation gives rise to a persistent activation state of JAK2 kinase activity by disrupting the autoinhibitory interaction between the pseudokinase domain (JH2) and the kinase domain (JH1). Activated JAK2 phosphorylates STAT 3/STAT 5, induces its dimerization and translocates to the nucleus where it binds to the gamma-interferon activation sites (GAS) in the promoter region of target genes [27]. STAT5 upregulates the expression of pro proliferative genes (such as *cyclin D1* and *MYC*) and anti-apoptotic genes (such as *BCL-XL* and *MCL-1*), driving cell cycle deregulation and apoptosis resistance, respectively. In PV, STAT5 cooperatively activates erythroid transcription factors (GATA1, klf1), leading to erythropoiesis; in PMF, STAT3 mediated secretion of IL-6 and TNF-α activates BM mesenchymal cells, promoting collagen deposition and fibrosis [28]. In addition, the cells with *JAK2* mutations transmit pro-inflammatory molecules such as *miR-155* (a key pro-inflammatory microRNA) by secreting exosomes, further amplifying microenvironmental inflammatory signals and forming a vicious cycle.

Dysregulation of intracellular signaling pathways represents the core effector mechanism translating genetic and epigenetic lesions into cellular phenotypes. Constitutive JAK-STAT activation remains the hallmark. Beyond this central axis, PI3K/AKT/mTOR signaling is robustly engaged. JAK2 phosphorylates insulin receptor substrate (IRS) proteins, recruiting the PI3K catalytic subunit p110. This catalyzes phosphatidylinositol (4,5)-bisphosphate (PIP2) conversion to phosphatidylinositol (3,4,5)-trisphosphate (PIP3), activating AKT. Activated AKT drives metabolic reprogramming via upregulation of glucose transporters (*GLUT1*) and hexokinases (*HK2*), enhancing glycolysis. Simultaneously, AKT activates the mTORC1 complex, stimulating protein synthesis through S6 kinase (S6K) phosphorylation. Crucially, mTORC1 also inhibits autophagy initiation by suppressing ULK1, causing accumulation of misfolded proteins (e.g., mutant CALR). This induces endoplasmic reticulum stress and genomic instability [29]. Hyperactivation of the PI3K/AKT/mTOR pathway clinically correlates with ruxolitinib resistance. Preclinical evidence suggests dual JAK2 and mTOR inhibition (e.g., with pacritinib) can reduce clonal burden and delay fibrosis progression [30].

RAS/MAPK/ERK pathway coactivation frequently occurs through crosstalk with JAK2-STAT5 or PI3K signaling. This induces GTP-loading of RAS proteins (KRAS/NRAS), triggering the RAF-MEK-ERK phosphorylation cascade. Activated ERK phosphorylates the ELK1 transcription factor, promoting formation of the activator protein 1 (AP-1) complex (Fos/Jun). AP-1 subsequently upregulates cell cycle regulators (e.g., cyclin E) and matrix metalloproteinases (MMPs), driving proliferation and facilitating extramedullary hematopoiesis [31]. *RAS* mutations (e.g., *NRAS G12D* mutation) substantially accelerate progression to acute myeloid leukemia (AML) by driving constitutive MAPK/PI3K signaling, often synergizing with *TP53* loss to induce blast accumulation. Single-cell sequencing studies confirm *RAS*-mutant subclones expand adaptively under therapeutic pressure via persistently active MAPK signaling, contributing to disease heterogeneity and relapse [32] (Figure 4).

Pathway interactions create a resilient network. STAT5 recruits DNA methyltransferase DNMT3A to tumor suppressor gene promoters (e.g., *SOCS3*), inducing their hypermethylation and silencing to relieve JAK2 negative feedback [33,34,35]. The PI3K/AKT axis activates hypoxia-inducible factor 1α (HIF-1α), which upregulates histone deacetylase HDAC2. HDAC2 compresses chromatin structure, repressing differentiation genes like *C/EBP α* [33,34,35]. RAS/MAPK signaling induces AP-1-dependent interleukin-8 (IL-8) secretion, recruiting myeloid-derived suppressor cells (MDSCs) to establish an immunosuppressive microenvironment. This network-level resilience explains why targeted therapies like JAK inhibitors alleviate symptoms but often fail to eradicate mutant clones [36].

### 2.5. Nondriver Mutations

Non-driver mutations significantly influence disease progression, phenotypic heterogeneity, and therapeutic resistance by cooperating with driver mutations. These mutations broadly classify into categories impacting epigenetic regulation, RNA splicing, and tumor suppression. Epigenetic regulator mutations include *TET2* (10–20% incidence), whose loss promotes DNA hypermethylation, silencing differentiation genes and tumor suppressors to synergize with *JAK2* mutations in PV development [37,38]. *ASXL1* mutations (~30% in PMF) disrupt PRC2 complex function, derepressing profibrotic and inflammatory genes to accelerate MF [21]. Spliceosome gene mutations (*SRSF2, U2AF1*; 10–15% prevalence) cause aberrant mRNA splicing of regulators controlling apoptosis or cell survival, conferring clonal fitness advantages and elevating leukemic transformation risk [39]. Tumor suppressor inactivation prominently features *TP53* mutations (occurring in 30–50% of MPN cases transforming to secondary AML). These mutations induce genomic instability and chemoresistance; co-occurrence with *JAK2 V617F* increases AML risk approximately tenfold [40].

Non-driver mutations follow a temporal hierarchy during clonal evolution. Early events, often involving *TET2* or *DNMT3A*, provide hematopoietic stem cells (HSCs) with a self-renewal advantage, creating a permissive context for subsequent driver mutations(*JAK2*, *CALR*) [41]. Intermediate-stage mutations, such as *ASXL1* or *SRSF2*, then impair hematopoietic differentiation or activate profibrotic signaling pathways, driving the phenotypic manifestations of MF or thrombocytosis [42]. Late events, including *TP53* or *RAS* mutations, disrupt DNA repair mechanisms or constitutively activate MAPK/PI3K signaling, ultimately triggering leukemic transformation [43].

Mechanistic cooperativity is evident: *ASXL1* mutations synergize with mutant *CALR* to amplify TGF-β signaling and fibrosis [25,44]; *TET2* deficiency cooperates with *JAK2 V617F* to suppress *PU.1* and block erythroid differentiation [24,45]; *SRSF2* mutations collaborate with STAT5 to hyperactivate PI3K/AKT signaling and upregulate the anti-apoptotic protein MCL-1 [41,46]; and *TP53* mutations exacerbate genomic instability and increase secretion of inflammatory cytokines (IL-6, TNF-α), intensifying BM microenvironment inflammation and fibrosis [47].Future research integrating single-cell multi-omics and clonal tracking is crucial to map these complex interactions and develop strategies targeting the malignant ecosystem [40].

## 3. Diagnosis of MPN

The diagnosis of MPN has undergone a significant transition from the Polycythemia Vera Study Group (PVSG) criteria to the World Health Organization (WHO) criteria [48]. The 2016 WHO revised criteria highlight the critical importance of morphological analysis of BM biopsy samples in the diagnosis and differential diagnosis of MPNs. BM pathology was elevated from a minor diagnostic criterion to a major diagnostic criterion. The WHO has revised the diagnostic criteria for MPN multiple times, with the 2022 revision showing minimal changes compared with the 2016 standard [48,49].

Techniques for molecular detection include allele-specific PCR (AS-PCR), quantitative PCR (qPCR) [50], Sanger sequencing and next-generation sequencing (NGS). With their rapid turnaround time and low cost, traditional methods such as AS-PCR and qPCR remain the first-line approaches for initial screening of *JAK2 V617F* and *CALR* mutations. Sanger sequencing is employed to detect unknown mutations in *JAK2* exon 12, *CALR*, and *MPL*, with high specificity.

NGS has assumed an increasingly critical role in the diagnosis of MPNs. One key advantage of NGS-based mutation profiling is its ability to simultaneously identify rare variants in *JAK2*, *CALR*, or *MPL* that might evade detection by conventional methods [51]. Additionally, whole-genome sequencing (WGS) and whole-exome sequencing (WES) have demonstrated growing potential in both research and clinical diagnostics. These approaches enable the discovery of novel mutations and noncoding genomic variations, providing critical insights for investigating complex cases and advancing scientific understanding of disease mechanisms.

In addition, clinical diagnosis includes laboratory examinations, such as blood cell count and blood biochemistry. Depending on the disease characteristics of different types of MPNs, clinicians employ relevant diagnostic methods.

### 3.1. PV

According to the 2022 WHO criteria, a diagnosis of PV can be established if a patient meets three major criteria or the first two major criteria plus the minor criterion [49,52]. In comparison to the 2008 criteria, the 2016 revision lowered the diagnostic thresholds for hemoglobin and hematocrit (Hct) levels. This adjustment primarily aimed to identify a newly recognized entity termed “masked PV” [48,53], which is associated with delayed diagnosis, insufficient therapeutic intensity and poorer prognosis.

The diagnosis of PV should be differentiated from reactive erythrocytosis. The condition is typically caused by chronic hypoxic states such as chronic lung disease, smoking or residing at high altitudes. Erythrocytosis in these cases represents an adaptive response to hypoxia and can be ruled out through detailed clinical history and auxiliary testing. In addition, certain hereditary disorders, such as *VHL* gene mutations or serum erythropoietin (EPO) receptor gene mutations, may lead to erythrocytosis. These patients usually have a clear family history and lack the *JAK2 V617F* mutation [54,55].

### 3.2. ET

According to the 2022 WHO classification [49], the diagnosis of ET requires meeting four major criteria or the first three major criteria plus the minor criterion [49,52]. The updated WHO classification placed greater emphasis on the molecular genetic testing and incorporated newly identified genetic mutation markers [56].

Diagnosis of ET requires careful differentiation from reactive thrombocytosis and other hematologic disorders associated with thrombocytosis, such as chronic myeloid leukemia (CML) with thrombocytosis. BM morphology plays a critical role in this differential diagnosis. Reactive thrombocytosis is characterized by an increased number of megakaryocytes. However, these cells do not show morphological atypia and are typically mature, occasionally displaying pleomorphic features [57].

Nearly all ET patients harbor the *JAK2 V617F* mutation, or in rare cases, approximately 3% of patients have *JAK2* exon 12 mutations. Therefore, in the absence of these mutations, secondary causes of thrombocytosis should be considered, prompting further investigation.

Reactive thrombocytosis typically involves identifiable underlying causes, and platelet counts often normalize after the triggering factor is addressed. For CML-related thrombocytosis, the presence of the *BCR-ABL* fusion gene serves as the key distinguishing feature [58].

### 3.3. MF

MF comprises two distinct entities: PMF and SMF. The latter category includes post-PV MF and post-ET MF [59].

#### 3.3.1. PMF

The 2016 WHO revision further subclassified PMF into two distinct subtypes: prefibrotic MF (Pre-MF) and overt MF [48].

##### Pre-MF

The diagnosis of pre-MF necessitates fulfillment of all three major criteria and at least one minor criterion, confirmed on two consecutive assessments [48,49,52].

The 2022 WHO classification maintains a molecular-driven framework, as the presence of additional mutations (*ASXL1*, *EZH2*, *IDH1/2*, *SRSF2*, and *U2AF1*) is associated with adverse prognosis and is termed “high-molecular-risk mutations” [49,60].

##### Overt-PMF

The diagnosis of overt MF requires the fulfillment of three major criteria and at least one minor criterion [49,52]. In the case of overt PMF, the 2016 WHO criteria retained the same major diagnostic criteria as the 2008 classification [48]. Notably, refinements focused on grading BMF, with updated histological thresholds to better define disease progression.

##### Differential Diagnosis

In the prefibrotic phase of PMF (pre-PMF), blood tests often reveal anemia, moderate leukocytosis, and/or thrombocytosis. Owing to the presence of thrombocytosis, distinguishing between pre-PMF and ET is crucial. Abnormal megakaryocytes in BM biopsy sample serve as a key diagnostic feature. In ET, megakaryocytes are distributed in loose clusters, appearing large and mature with a regular nuclear morphology [61,62,63]. In contrast, pre-PMF is characterized by tight clusters of megakaryocytes that display abnormal maturation with hyperchromatic and irregularly folded nuclei.

Monocytosis (≥1 × 10^9^/L) is a hallmark of disease progression in PMF. When PB monocyte counts exceed 1 × 10^9^/L, chronic myelomonocytic leukemia (CMML) becomes a diagnostic consideration [64]. Molecular profiling for MPN driver mutations should be performed to avoid misdiagnosis of chronic myelomonocytic leukemia (CMML), which is typically associated with mutations in genes such as *ASXL1*, *TET2*, or *SRSF2*.

#### 3.3.2. SMF

SMFs are categorized into two subtypes, post-PV MF and post-ET MF. Diagnosis is primarily based on criteria from the ICC and the WHO [49]. The 2022 ICC outlines diagnostic requirements for post-PV MF and post-ET MF, necessitating the fulfillment of two major criteria and at least two minor criteria [59]. Molecular genetic evaluation is pivotal. Molecular genetic profiling constitutes a cornerstone of diagnostic evaluation, with all post-PV MF cases demonstrating *JAK2* mutations. While approximately 50% of post-ET MF patients exhibit *JAK2 V617F* mutations, 30% carry *CALR* mutations, and 5–10% present with *MPL* mutations or are classified as triple-negative, which means a lack of *JAK2*, *CALR*, or *MPL*-driven mutations. Furthermore, approximately 80% of SMF patients harbor additional somatic mutations in genes such as *ASXL1* and *TET2* [65]. The diagnosis of SMF requires exclusion of other MPNs or secondary conditions, supported by integrated evaluation of clinical, morphological, and molecular features for precise stratification.

## 4. Prognosis of MPN

Once a patient is diagnosed, a judgment should be made regarding the patient’s prognostic grouping to better guide treatment choices.

### 4.1. Prognosis of PV

#### 4.1.1. Thrombus Risk Stratification

Stratification of patients according to thrombotic history and age categorizes individuals into high-risk and low-risk groups: ① High-risk group: Age ≥ 65 years and/or previous PV-related arterial or venous thrombosis. ② Low-risk group: Age < 65 years and/or no previous PV-related arterial or venous thrombosis [66,67].

#### 4.1.2. Survival Prognosis Grouping

For patients without NGS testing, the International Working Group for PV (IWG-PV) divides patients into low-risk (0 points), intermediate-risk (one or 2 points) and high-risk (≥3 points) categories on the base of age, leukocyte count and venous thrombosis. The median OS times are 28, 19, and 11 years, respectively [68].

For patients with NGS testing, an enhanced system with added gene mutations can be adopted, which categorizing patients as low (0 or 1 point), intermediate (2 or 3 points), or high (≥4 points) clinical risk. The median OS times are 24.0, 13.1, and 3.2 years, respectively [68].

#### 4.1.3. Post-PV Survival Prognosis Grouping

The MF secondary to PV and ET (MYSEC-PM) prognostic model is adopted for post-PV patients and classifies patients into low-risk (<11 points), medium-risk 1 (≥11 points), medium-risk 2 (14 points to <16 points), and high-risk (≥16 points). The median survival time was not reached in the low-risk group, 9.3 years in the moderate 1-risk group, 4.4 years in the moderate 2-risk group and 2 years in the high-risk group [69].

### 4.2. Prognosis of ET

#### 4.2.1. Overall Survival Prognosis Judgment

The leukemia transformation rate of ET patients 20 years after diagnosis is less than 5%, and the progression rate of MF is slightly higher [3]. Mayo Clinic data indicate that ET patients survive 18 years on average, while the median survival time reaches 33 years in those aged below 60 years. However, their life expectancy remains below the matched population [56,68]. In 2012, the International Working Group for MF Research and Treatment proposed the International Prognostic Scoring System for ET (IPSET). Patients are divided into three prognostic categories: low-risk (0 points), intermediate-risk (1–2 points), and high-risk groups (≥3 points), with corresponding median survival periods of unreached, 24.5 years, 13.8 years [70].

With the development of NGS technology, studies have found that gene mutations are also risk factors affecting survival. A multicenter retrospective study involving 1607 ET patients revealed that patients with a *JAK2 V617F* mutation (load > 35%), *CALR* type I mutation (or similar type I mutation), or *MPL* mutation have a higher risk of progressing to MF [71]. Recently, the mutation-enhanced international prognostic systems (MIPSS) model was proposed concerning gene mutation scoring [72].

#### 4.2.2. Thrombotic Risk Prognosis Judgment

ET prognosis closely links to thrombosis risk. The risk factors for arterial thrombosis include age > 60 years, cardiovascular risk factors (CVF, including diabetes, hypertension, hypercholesterolemia, or smoking), previous history of thrombosis, *JAK2 V617F* mutation, and white blood cell count at least 11 × 10^9^/L [73,74,75,76]. When the platelet count is greater than 1000 × 10^9^/L, the risk is decreased [75,77]. The risk factors for venous thrombosis include male sex, age > 60 years, previous history of thrombosis, and positive *JAK2 V617F* mutation [75]. In 2012, Italian scholars proposed the International Prognostic Score for Thrombosis in ET (IPSET-thrombosis). Patients are stratified into low-risk group (0–1 point), medium-risk group (2 points), and high-risk group (≥3 points), and annual thrombosis incidence rates of them are 1.03%, 2.35%, and 3.56%, respectively [76].

Later, Barbui et al. proposed the revised IPSET-thrombosis, in which ET was classified into the very low-risk group (without thrombosis history, age ≤ 60 years, and negative *JAK2 V617F* mutation), the low-risk group (without thrombosis history, age ≤ 60 years, and positive *JAK2 V617F* mutation), the intermediate-risk group (without thrombosis history, age > 60 years, and negative *JAK2 V617F* mutation), and the high-risk group (a history of thrombosis, or age > 60 years and positive *JAK2 V617F* mutation) [74]. This model guides ET treatment globally. Chinese scholar Fu Rongfeng’s team has also confirmed this model and proposed a revised IPSET suitable for Chinese patients [78].

Neither original nor revised IPSET-thrombosis scoring system demonstrates predictive value for venous thrombotic events [79].

### 4.3. Prognosis of PMF

The prognostic evaluation of PMF involves multiple scoring systems, including International Prognostic Scoring System (IPSS), Dynamic International Prognostic Scoring System (DIPSS), DIPSS-plus, Genetically Inspired Prognostic Scoring System (GIPSS), Mutation-Enhanced International Prognostic Scoring System for patients under 70 years (MIPSS70), MIPSS70-plus, MYSEC-PM, and MTSS (for patients undergoing HSCT).

IPSS was the first established system for prognostic assessment in PMF and has been validated solely for initial diagnosis [80,81]. It evaluates prognosis based primarily on clinical features and laboratory data, including age, hemoglobin level, white blood cell count, PB blast percentage, and the presence of constitutional symptoms [82].

DIPSS, developed from IPSS but allocating two points to hemoglobin levels below 10 g/dL, is valid for the entire disease course, from initial diagnosis to disease progression [83]. This adjustment led to the establishment of four distinct prognostic categories: low risk (0 points), intermediate-1 risk (1–2 points), intermediate-2 risk (3–4 points), high risk (5–6 points) [83]. Furthermore, DIPSS-Plus further refines DIPSS by adding additional points derived from platelet count, red blood cell transfusion dependency, and the presence of an adverse karyotype [84]. Implementation of the DIPSS-Plus model involves two sequential steps: ① determination of the preliminary DIPSS classification (scored 0–3 corresponding to low through high-risk categories); ② cumulative addition of 1-point increments for three clinical conditions: platelets < 100 × 10^9^/L, erythrocyte transfusion requirement, and unfavorable karyotype [81,84].

In contemporary prognostic research, mutations have been incorporated into the formulation of the following novel risk stratification systems.

MIPSS70 integrates clinical conditions, cytogenetic findings, and mutational status specifically for PMF patients within transplant eligibility criteria (age ≤ 70 years). This model incorporates nine parameters, consisting of three molecular markers and six clinical indicators [85].

MIPSS70-Plus version 2.0 improves upon MIPSS70. This updated model comprises five genetic variables and four clinical manifestations, enhanced incorporation of cytogenetic data. Notably, it excludes BMF grading along with leukocyte and platelet count parameters and incorporated three key modifications: integration of very-high-risk karyotype, reclassification of *U2AF1 Q157* variant as an high-molecular-risk mutation, and establishment of sex-specific hemoglobin cutoff values with severity [86].

A multicenter cohort study conducted by researchers from Mayo Clinic and the University of Florence evaluated 641 patients with PMF. Through genetic-focused multivariable regression modeling, very-high-risk karyotype (2 points), unfavorable karyotype (one point), absence of type 1/like *CALR* mutation (one point) and presence of *ASXL1* (one point) *SRSF2* (one point) and *U2AF1Q157* (one point) mutations are identified as independent risk factors for survival [87]. This system stratifies PMF patients by genetic variants alone, thus called GIPSS (Table 1), which was further validated in 266 patients diagnosed with PMF or post-ET/post-PV MF and appeared to be superior to DIPSS in instances where the two models were discordant [88].

Figure 5 shows how risk stratification guides clinical treatment.

## 5. Treatment Modalities for MPN

In this section, the treatment strategies are discussed for MPNs: PV, ET, PMF, and post-PV MF and post-ET MF.

### 5.1. PV

PV is a relatively benign and slow-progressing MPN [89]. Theoretically, the treatment of PV mainly targets four goals: controlling erythrocytosis, reducing blood viscosity, preventing thrombosis and hemorrhagic complications, and delaying the progression to MF or acute leukemia. Nevertheless, no current drug therapy has proven to prolong life or halt transformation to MF or AML. Consequently, contemporary therapeutic strategies focus prioritize minimizing thrombotic risk [53].

#### 5.1.1. Traditional Treatment

First-line management typically combines phlebotomy, low-dose aspirin, and cytoreductive therapy (mainly for high-risk individuals) [90,91].

Phlebotomy is indicated for almost all PV patients. Regular blood removal (250–500 mL daily, every other day, or twice weekly) reduce Hct to 45%, alleviating symptoms and thrombosis risk [92,93]. However, repeated bloodlet may cause side effects such as iron deficiency anemia. Therefore, for weak patients, the amount of bloodlet should not exceed 250 ml [89].

Low-dose aspirin (81 mg/day) is added unless contraindicated, providing antithrombotic effects and microvascular symptom relief [94,95]. Notably, its use requires evaluation for acquired von Willebrand syndrome (AvWS) in extreme thrombocytosis (platelets ≥ 1000 × 10^9^/L) [96,97,98].

Cytoreductive agents (such as hydroxyurea or IFN-α) are reserved for high-risk patients or those who need long-term control [99,100]. These inhibit ribonucleotide reductase, reducing the blood cell count [101,102]. Notably, leukocyte monitoring is essential to mitigate acute leukemia risk [100]. Intriguingly, some researchers have proposed cytoreduction may benefit select low-risk patients by controlling symptoms and reducing phlebotomy frequency-though not for thrombosis prevention [103,104,105,106,107,108]. Among these agents, IFN-α (including pegylated forms) has been a cornerstone for >35 years, particularly in PV, due to its anti-proliferative effects and low leukemogenic risk [109]. Owing to its anti-proliferative effect and low risk of leukemia, it is usually suitable for young patients or those who are intolerant to hydroxyurea [103,107,110]. Significantly, ropeginterferon alfa-2b (Ropeg), a novel pegylated IFN-α, reduces *JAK2 V617F* allele burden and exhibits a superior safety profile versus hydroxyurea or other IFN formulations [111].

Second-line options usually include: JAK2 inhibitors (such as ruxolitinib) are usually used for hydroxyurea-resistant/intolerant patients, especially with severe pruritus or splenomegaly. Ruxolitinib can relieve the disease by reducing the *JAK2 V617F* allele burden [112], lower Hct, and provide durable responses [113,114].

For refractory advanced PV patients, oral busulfan can also be considered, which is helpful for hematological remission, but the potential risk of leukemia caused by it should still be considered [68,115,116,117,118].

#### 5.1.2. New Treatments

Expanding beyond cytoreduction, immune modulation represents an emerging frontier. Ruchi Yadav proposed that blocking T-cell inhibitory pathways in the tumor microenvironment (TME) via next-generation immune checkpoint inhibitors (ICIs) could exert antitumor effects in PV [119,120].

### 5.2. ET

Unlike PV, the therapeutic strategies for ET are predominantly predicated on a meticulous risk stratification system that focuses on vascular stratification. This comprehensive risk assessment framework categorizes patients into distinct groups—very low-risk, low-risk, moderate-risk, and high-risk—based on age, vascular event history, and *JAK2* mutations status, guiding personalizes therapy [121,122,123,124].

#### 5.2.1. Risk-Adapted Therapy

For very low-risk and low-risk patients predominantly focus on antiplatelet therapy. Low-dose aspirin is frequently incorporated into treatment regimens, primarily to alleviate microvascular manifestations [125]. Caution is warranted if platelet counts surge beyond 1000 × 10^9^/L, the use of aspirin demands meticulous consideration [124].

For moderate-risk patients, the therapeutic strategy typically encompasses low-dose aspirin monotherapy or a combination of hydroxyurea and aspirin. Cytoreduction may be considered for *CALR/MPL*-mutated cases [124].

In the high-risk patient cohort, a regimen of twice-daily aspirin administration has emerged as an experimental approach. Additionally, evidence from clinical studies underscores proton pump inhibitors should accompany aspirin in elderly patients (>70 years) to prevent gastric injury [124,126,127].

#### 5.2.2. Cytoreductive Treatment

Consistent with PV principles, hydroxyurea is firmly established as the first-line cytoreductive agent, whereas anagrelide serves as a second-line alternative. Although anagrelide is associated with an elevated risk profile, including arterial thrombosis, bleeding events, and MF compared with hydroxyurea, it has a notable advantage in reducing the incidence of venous thrombosis [128,129,130,131]. Similarly to PV, interferons (such as pegylated (PEG) interferon α) have also demonstrated efficacy in maintaining optimal platelet count control, further enriching the cytoreductive treatment options for ET [132].

#### 5.2.3. Targeted Therapies

Growing interest in ET targeted therapies is driving research into its complex interplay with the tumor microenvironment (TME). Leading groups especially identify T-cell modulation and immune checkpoint inhibitors as potential breakthrough pathways [119].

Scientific investigation has established that lysine-specific demethylase (LSD1) regulates malignant stem cell self-renewal and macrophage maturation. With respect to this mechanism, the lysine-specific demethylase (LSD1) inhibitor IMG-7289 (Bomedemstat) has entered clinical trials for the treatment of ET. Clinical data indicate IMG-7289 not only stabilizes hemoglobin levels more effectively than ruxolitinib, but also significantly reduces driver gene mutations like *JAK2V617F* [133,134]. Notably, telomerase inhibitors have achieved favorable hematological response rates and molecular remission in patients receiving ET treatment. However, their clinical utility is limited by a higher frequency of adverse events, including myelosuppression and gastrointestinal toxicity. Additionally, therapies targeting the MDM2-p53 signaling pathway have garnered increasing interest [135]. MDM2 inhibitors function by blocking p53 degradation, reactivating its anticancer effects, and thus representing promising alternatives for ET management [136,137].

### 5.3. MF

#### 5.3.1. Conventional Management

Currently, allogeneic HSCT represents the only potentially curative therapy for MF, with clinical evidence demonstrating significant improvement in overall survival [138,139,140].

For MF patients with splenomegaly, preoperative measures for allogeneic HSCT typically involve splenectomy, the administration of JAK inhibitors, or involved-field radiotherapy [141,142,143]. Research indicates that pre-HSCT ruxolitinib accelerates engraftment of leukocytes and improves post-transplant outcomes [144,145].

For non-transplant candidates, in cases where MF patients have anemia without splenomegaly, first-line treatment includes non-JAKi drugs such as testosterone enanthate (administered via intramuscular injection at a weekly dose of 400–600 mg), prednisolone (at a daily dose of 0.5 g/kg), and danazol (at a daily dose of 600 mL), though hepatotoxicity and masculinization are concerns [146,147].

For MF patients without anemia, hydroxyurea is commonly used as the first-line drug. It can effectively reduce the splenic size but may cause side effects such as myelosuppression [148]. Like PV and ET, when patients are intolerant to or refractory to hydroxyurea, ruxolitinib can be used [80,149]. For patients with drug-refractory MF manifesting splenic abdominal pain, symptomatic portal hypertension, severe thrombocytopenia, or transfusion requirements, splenectomy is often the preferred therapeutic approach. Moreover, postoperative prevention of thrombosis formation is essential [150,151].

Evidence from recent studies supports low-dose JAK inhibitors (momelotinib and pacritinib) are second-line therapeutic strategies, alleviating splenomegaly, improving anemia, and reliving symptoms [80]. Momelotinib is first-line for patients with anemia and splenomegaly, despite peripheral neuropathy risks [80,152].

#### 5.3.2. Novel Immunotherapies and Targeted Drugs

Novel immunotherapies and targeted drugs are offering new hope to patients with MF. Preclinical studies indicate that immune checkpoint inhibitors, including those targeting the PD-1/PD-L1 axis, show promising potential. This potential reactivates the immune system, enabling more effective elimination of abnormal myeloid cells. Notwithstanding encouraging preclinical evidence, clinical translation remains challenged by unresolved issues. These issues include selecting patients likely to benefit, managing potential immune-related adverse events, and exploring optimal combinations with other therapies like chemotherapy or targeted drugs to improve outcomes [153].

Accumulating evidence positions MPNs as inflammation-driven disorders, with cytokine imbalances impacting prognosis [154]. While chronic inflammation is recognized as a key driver of MPN development, the specific functional roles of stem cell-intrinsic inflammasome signaling in the pathogenesis of the disease remain inadequately understood. This knowledge gap constitutes a significant avenue for future research [155]. Relevant biomarkers include elevated S100A proteins in granulocytes/plasma and pan-hematopoietic NF-κB pathway activation [156,157,158]. Promisingly, the IRAK4 inhibitor CA-4948 reduces disease burden and inflammatory cytokines in MPNs, highlighting this pathway’s therapeutic relevance [159].

Among numerous novel therapeutic strategies, rusfertide has demonstrated guiding significance for controlling erythrocytosis in PV patients. By mimicking hepcidin, it inhibits cellular iron transport into the bloodstream, thereby suppressing erythropoiesis and achieving the goal of controlling red blood cell mass [160]. Notably, building on the understanding that restoring p53 function via modulation of the pro-apoptotic Bcl-2 protein family can induce apoptosis in MF tumor cells, navtemadlin (KRT-232) has been proposed as a novel therapeutic approach for MF. It acts through inhibition of murine double minute 2 (MDM2), leading to reactivation of p53 function [161]. Furthermore, the combination of navitoclax with JAK1/2 inhibitors, including ruxolitinib, significantly enhances the efficacy of ruxolitinib even in ruxolitinib-resistant cells [162]. Such combination therapies have shown promising potential in effectively managing clinical sequelae, underscoring their considerable clinical relevance [163] (Table 2).

## 6. Conclusions 

This review synthesizes advances in MPN molecular pathogenesis and therapeutics, updating diagnostic and prognostic standards. The integration of structural, functional, and molecular imaging—spanning invasive BM biopsies to noninvasive techniques—has enhanced early detection sensitivity and specificity. Emerging therapies targeting epigenetic dysregulation and immune microenvironment interactions demonstrate superior efficacy to frontline treatments in preclinical/clinical settings, yet face translational challenges including patient selection and significant toxicity. While stem cell-intrinsic inflammasome signaling remains under investigation at the pathogenesis level, combination therapies and mutation-independent immunotherapies represent promising frontiers. Deciphering the complexities of these illnesses and enhancing patient outcomes require ongoing study and cooperation.

## Figures and Tables

**Figure 1 cancers-17-03142-f001:**
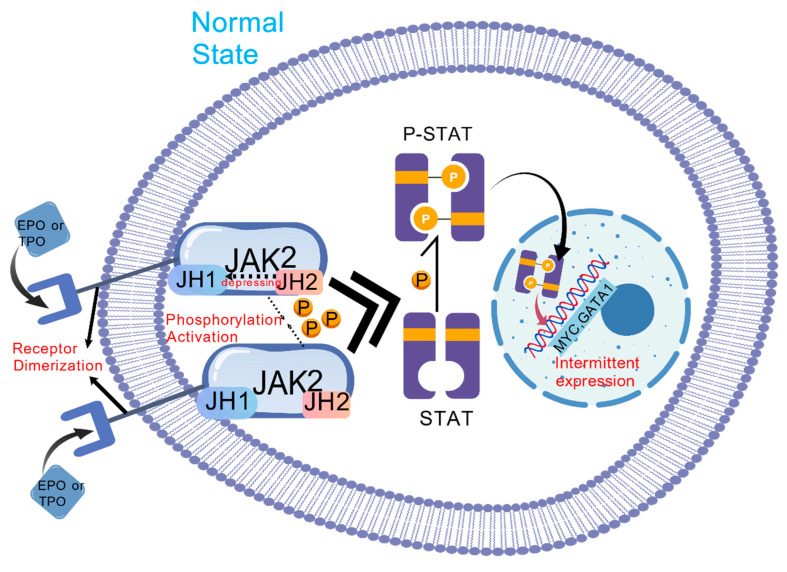
Expression of *JAK2* under normal conditions. In the normal JAK-STAT activation pathway, the JH2 pseudokinase domain of JAK can inhibit the JH1 catalytic domain through hydrophobic interaction and hydrogen bonding. When the extracellular ligand (EPO/TPO) binds to the transmembrane receptor, the receptor dimerizes, leading to the proximity of adjacent JAK2 molecules to each other and phosphorylation activation, followed by phosphorylation of STAT, so that it forms a dimer and enters the nucleus, binds to the gas sequence on the DNA, and activates Pro proliferation/differentiation genes (such as *MYC*, *GATA1*).

**Figure 2 cancers-17-03142-f002:**
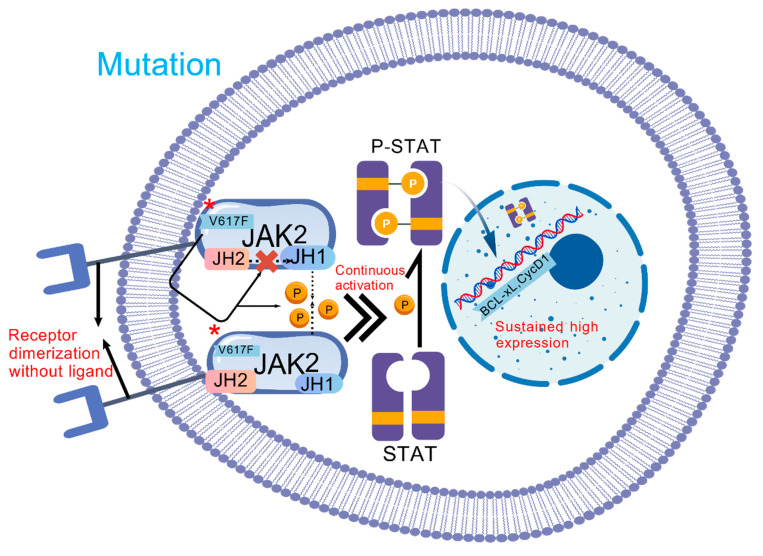
The effect of the *JAK2 V617F* mutation on MPNs. In the signaling pathway under the *V617F* mutation, the collapse of the hydrophobic interface separates JH1 from JH2, and the *V617F* mutation disrupts JH2-JH1 autoinhibition, so that JAK2 can still be continuously activated through receptor dimerization without ligand binding, catalyzing STAT phosphorylation, and a large number of P-STAT dimers enter the nucleus, so that the target genes (such as *BCL XL*, *CYCLIN D1*) are continuously highly expressed. Asterisk represents mutation.

**Figure 3 cancers-17-03142-f003:**
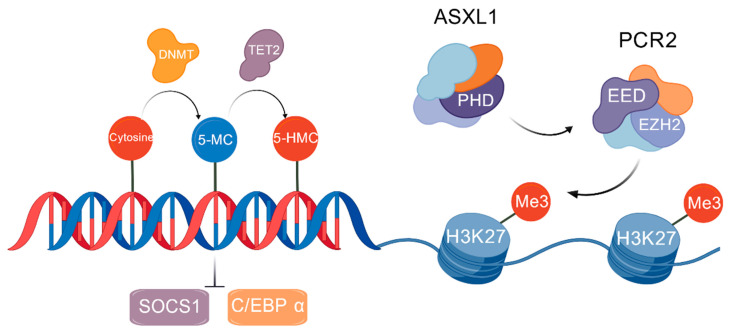
The mechanisms of epigenetic changes in MPNs. DNA methyltransferase (DNMT) catalyzes cytosine in GpC island to 5-methylcytosine (5-MC), whereas ten-eleven translocation 2 (TET2) catalyzes 5-MC to form 5-hydroxymethylcytosine (5-HMC), leading to demethylation, which causes expression of *SOCS1* and *C/EBP α.* Additional sex combs like 1 (ASXL1) through its PHD domain interacts with polycomb repressive complex 2 (PRC2) which catalyzes trimethylation (Me3) of H3 on lysine 27. *ASXL1* mutations (30% in PMF) truncate its PHD domain and disrupt the interaction with the PRC2, resulting in reduced the level of trimethylation of H3 on lysine 27.

**Figure 4 cancers-17-03142-f004:**
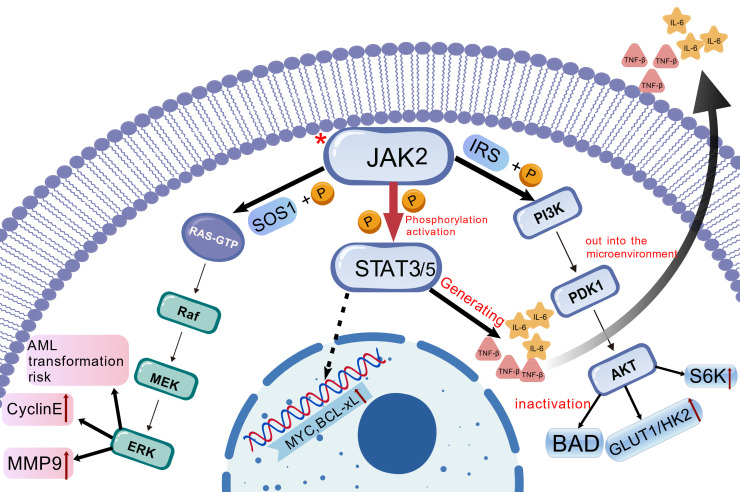
Three dysregulated signaling pathways in MPNs: JAK-STAT, PI3K/AKT/mTOR and MAPK/ERK. Mutant JAK2 persistently activates STAT3/5. STAT5 upregulates the expressions of pro proliferative genes (such as *cyclin D1* and *MYC*) and anti-apoptotic genes (such as *BCL-XL* and *MCL-1*). The PI3K/AKT/mTOR pathway synergizes with JAK-STAT signaling. JAK2 phosphorylates the insulin receptor substrate (IRS) protein, recruits the PI3K, catalyzes phosphatidylinositol diphosphate (PIP2) to generate inositol triphosphate (PIP3), and activates 3-phosphoinositide-dependent protein kinase 1 (PDK1) and AKT. Activated AKT drives glycolysis (upregulation of *GLUT1* and *HK2*) and protein synthesis (activation of S6K) by phosphorylating the pro apoptotic protein BAD. JAK2 phosphorylates SOS1 and promotes RAS (KRAS/NRAS) to bind GTP, which triggers RAF-MEK-ERK cascade. Asterisk represents mutation.

**Figure 5 cancers-17-03142-f005:**
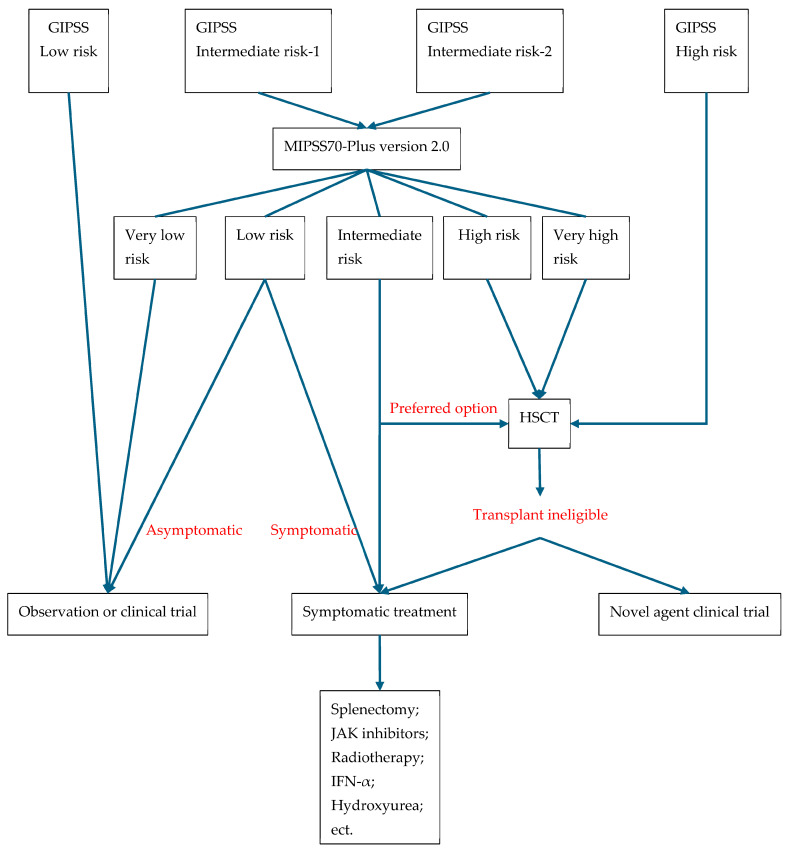
Risk-adapted treatment for MF.

**Table 1 cancers-17-03142-t001:** GIPSS [87].

Risk Factor	Score	Risk Level	Median OS/Years
Very-high-risk karyotype ^a^	2	Low risk: 0 point	26.4
Unfavorable karyotype ^b^	1	Intermediate risk-1: 1 point	8
Absence of *CALR* type 1-like mutation	1	Intermediate risk-2: 2 points	4.2
*ASXL1* mutation	1	High risk: 3 or more points	2
*SRSF2* mutation	1		
*U2AF2 Q157* mutation	1		

^a^: Including *ASXL1, EZH2, SRSF2, U2AF1 Q157*, and *IDH1/2*. ^b^: complex karyotype or one or two abnormalities that include trisomy 8, del 7/7q, i(17q), del 5/5q, del 12p, inv(3), or 11q23 rearrangement.

**Table 2 cancers-17-03142-t002:** New therapies about PV and MF.

	Therapeutic Agent	Mechanism of Action	Clinical Trial Status
PV	Rusfertide	Inhibits iron transport into the bloodstream, thereby suppressing erythropoiesis	International Phase II REVIVE trial
MF	navtemadlin (KRT-232)	Inhibits murine double minute 2 (MDM2), leading to reactivation of p53 tumor suppressor function	Phase III BOREAS trial
	Navitoclax	Functions as a BCL-2 homology 3 (BH3) mimetic; binds to pro-survival BCL-2 family proteins, disrupts their interaction with pro-apoptotic factors, and promotes apoptosis of malignant MF cells	Phase II REFINE trial

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
