# Peer review of "Advances in the Diagnosis and Treatment of Myeloproliferative Neoplasms (MPNs)"

_cancers, 2025, doi:10.3390/cancers17193142_

Round 1
Reviewer 1 Report
Comments and Suggestions for Authors
In this review, Authors summarizes the latest advancements in the diagnosis and treatment of myeloproliferative neoplasms (MPNs), focused on the importance of molecular mechanisms in guiding therapeutic approaches and the potential for precision medicine in the future.
The study is valuable as it addresses a very important and current topic. The study has been written in accordance with scientific regulations. In my opinion, the manuscript should be accepted for publication after minor revision. In the manuscript there are some editing errors, such as uniform font. Figures should be improved. The quality of the figures is poor, they are difficult to read.
Author Response
Comment: In the manuscript there are some editing errors, such as uniform font. Figures should be improved. The quality of the figures is poor, they are difficult to read.
Response: We sincerely apologize for our carelessness and lack of rigor. We have unified the format and modified the pictures. The original Figure 1 has been split into two pictures (Figure 1 and 2), which describe the normal and abnormal JAK - STAT pathways respectively (page5, line145-159). We have adjusted the original Figure 2 (now Figure 3) to make it more concise and ensure that its color is consistent with the other three figures (page7, line203). We believe that these modifications will make figures easier to understand.

Reviewer 2 Report
Comments and Suggestions for Authors
This is a well-structured review article that provides an overview of the recent advances in the biology and clinical management of myeloproliferative neoplasms (MPNs). The authors have synthesized a large body of literature, covering the critical areas of genetic mutations, risk stratification, diagnosis, and treatment.
Suggestions:
- The depth on novel therapeutic agents: While the treatment section covers established therapies (e.g., hydroxyurea, ruxolitinib, interferon), it could be enhanced by a more detailed discussion of the newer, promising agents currently in advanced clinical trials. Please ensure latest clinical trial updates from ASH or EHA are included. For instance, expanding on the mechanisms, latest phase II/III data, and potential place in therapy for agents like: Navitoclax and other BCL-XL inhibitors in combination with ruxolitinib, PVRK2 inhibitors (e.g., rusfertide) for polycythemia vera, MDM2 inhibitors (e.g., KRT-232) for TP53 mutated patients, BET inhibitors and other novel epigenetic modifiers. A dedicated paragraph or table summarizing these new therapies would greatly increase the review's impact and forward-looking perspective.
- Integration of molecular data into prognostication: The section on risk stratification correctly highlights the importance of molecular profiling. However, this could be strengthened by more explicitly discussing how specific mutation combinations (e.g., JAK2 vs. CALR with additional ASXL1 or TP53 mutations) concretely influence clinical decision-making beyond the standard IPSS and MIPSS scoring systems. A flow-chart or a clear algorithmic summary for risk-adapted therapy selection would be helpful for the reader.
- Clonal Evolution and Pre-MPN States: The review touches upon genetics but could briefly explore the emerging concepts of clonal hematopoiesis (CHIP) and how these precursor states evolve into overt MPNs. A short paragraph on this topic would provide a more complete molecular pathogenesis narrative.
- Figures: Please ensure the style, color and size of words in Figures are consistent. Please ensure the illustrated pathway is easily understood.
- References: Some references are incomplete or formatted inconsistently (e.g., Ref 1, 6, 9, 148, 150, etc lack page numbers). Please ensure all references follow the journal‘s style guide during revision.
Conclusion:
This review summarizes a complex and rapidly evolving field with clarity. It is suggested to further enhance its depth, clinical utility, and educational value. I believe this manuscript will be a valuable contribution to the literature upon the incorporation of these changes.
Author Response
Comments 1: The depth on novel therapeutic agents: While the treatment section covers established therapies (e.g., hydroxyurea, ruxolitinib, interferon), it could be enhanced by a more detailed discussion of the newer, promising agents currently in advanced clinical trials. Please ensure latest clinical trial updates from ASH or EHA are included. For instance, expanding on the mechanisms, latest phase II/III data, and potential place in therapy for agents like: Navitoclax and other BCL-XL inhibitors in combination with ruxolitinib, PVRK2 inhibitors (e.g., rusfertide) for polycythemia vera, MDM2 inhibitors (e.g., KRT-232) for TP53 mutated patients, BET inhibitors and other novel epigenetic modifiers. A dedicated paragraph or table summarizing these new therapies would greatly increase the review's impact and forward-looking perspective.
Response 1: We appreciate your suggestion regarding the need for more information on novel therapeutic agents. We have summarized the mechanisms and the latest clinical trial data of Rusfertide, navtemadlin (KRT - 232), and Navitoclax in Table 2(page19-20, line662-675). We hope that the revised version meets your expectations.
Comments 2: Integration of molecular data into prognostication: The section on risk stratification correctly highlights the importance of molecular profiling. However, this could be strengthened by more explicitly discussing how specific mutation combinations (e.g., JAK2 vs. CALR with additional ASXL1 or TP53 mutations) concretely influence clinical decision-making beyond the standard IPSS and MIPSS scoring systems. A flow-chart or a clear algorithmic summary for risk-adapted therapy selection would be helpful for the reader.
Response 2: We think this is an excellent suggestion. We have draw a flow chart (Figure 5) to illustrate how risk stratification guides clinical treatment (page16, line524).
Comments 3: Clonal Evolution and Pre-MPN States: The review touches upon genetics but could briefly explore the emerging concepts of clonal hematopoiesis (CHIP) and how these precursor states evolve into overt MPNs. A short paragraph on this topic would provide a more complete molecular pathogenesis narrative.
Response 3: We are very grateful for your suggestions on molecular pathogenesis. We have added a description of CHIP to make the molecular mechanism more complete (page3, line96-116).
Comments 4: Figures: Please ensure the style, color and size of words in Figures are consistent. Please ensure the illustrated pathway is easily understood.
Response 4: We sincerely apologize for our lack of rigor. We have modified the pictures. The original Figure 1 has been split into two pictures (Figure 1 and 2), which describe the normal and abnormal JAK - STAT pathways respectively (page5, line145-159). We have adjusted the original Figure 2 (now Figure 3) to make it more concise and ensure that its color is consistent with the other three figures (page7, line203). We believe that these modifications will make figures easier to understand.
Comments 5: References: Some references are incomplete or formatted inconsistently (e.g., Ref 1, 6, 9, 148, 150, etc lack page numbers). Please ensure all references follow the journal’s style guide during revision.
Response 5: Thanks for your careful checks. We have improved the format of references 1,6,9,15,18,20,21,32,34,37,41,46,50,58,114,148,150 to ensure they are consistent with others, and we ensure all references follow the journal’s style guide. Once again, we appreciate your time and effort in reviewing our manuscript.
Reviewer 3 Report
Comments and Suggestions for Authors
The text clearly lists the diagnostic criteria; however, for a better overview, it would be beneficial to present them in a table.
Figures representing molecular mechanisms of pathogenesis are appropriate, clear, and informative. The addition of schematic summaries of prognosis and therapeutic pathways could further enhance readability and impact.
Author Response
Comments 1: The text clearly lists the diagnostic criteria; however, for a better overview, it would be beneficial to present them in a table.
Response 1: The suggestion you put forward is excellent. The manuscript we initially submitted included a table of diagnostic criteria, but we removed it due to copyright issues.
Comments 2: The addition of schematic summaries of prognosis and therapeutic pathways could further enhance readability and impact.
Response 2: We sincerely thank the reviewer for this comment. In the prognosis section, we have added Table 1 and Figure 5 to illustrate how different risk stratifications guide clinical treatment (page15-16, line519-524). In the diagnosis section, we have added Table 2 to summarize the latest treatment methods (page19-20, line662-675). We hope these revisions can improve the readability.